# Application of Artificial Neural Network to the Prediction of Tensile Properties in High-Strength Low-Carbon Bainitic Steels

**Sang-In Lee, Seung-Hyeok Shin and Byoungchul Hwang \***

Department of Materials Science and Engineering, Seoul National University of Science and Technology, Seoul 01811, Korea; sangin19@kist.re.kr (S.-I.L.); seunghyeok527@seoultech.ac.kr (S.-H.S.)
\* Correspondence: bhwang@seoultech.ac.kr; Tel.: +82-2-970-6638

**Abstract:** An artificial neural network (ANN) model was designed to predict the tensile properties in high-strength, low-carbon bainitic steels with a focus on the fraction of constituents such as PF (polygonal ferrite), AF (acicular ferrite), GB (granular bainite), and BF (bainitic ferrite). The input parameters of the model were the fraction of constituents, while the output parameters of the model were composed of the yield strength, yield-to-tensile ratio, and uniform elongation. The ANN model to predict the tensile properties exhibited a higher accuracy than the multi linear regression (MLR) model. According to the average index of the relative importance for the input parameters, the yield strength, yield-to-tensile ratio, and uniform elongation could be effectively improved by increasing the fraction of AF, bainitic microstructures (AF, GB, and BF), and PF, respectively, in terms of the work hardening and dislocation slip behavior depending on their microstructural characteristics such as grain size and dislocation density. The ANN model is expected to provide a clearer understanding of the complex relationships between constituent fraction and tensile properties in high-strength, low-carbon bainitic steels.

**Keywords:** artificial neural network; deformability; yield-to-tensile ratio; uniform elongation; bainitic steel

## 1. Introduction

Over the past several decades, crude oil and natural gas have undergone significant depletion owing to the increase in energy consumption by the rapid development in industrial technology. With the increases in oil drilling and transportation in extremely cold environments such as Alaska and Siberia, pipeline steels require better combinations of high strength and low-temperature toughness as well as good deformability [1–7]. Precise metallurgical designs comprising detailed chemistry control and advanced thermomechanical processing have been indispensable to achieve an excellent balance of mechanical properties, as an increase in the strength is generally accompanied by deteriorated toughness and ductility [5–20]. In particular, excellent deformability characteristics such as continuous yielding behavior, low yield-to-tensile ratio, and high uniform elongation have been increasingly required to improve the fracture and buckling resistance as opposed to gradual or sudden deformation caused by combined installation stress, external pressure, or ground movements in severe environments [15–20]. Because recently developed high-strength pipeline steels fabricated by heavy rolling reduction and accelerated cooling have highly complex bainitic microstructures, the correlation between the microstructures and tensile properties of the high-strength, low-carbon bainitic steels is more difficult to clearly understand.

Recently, an artificial neural network (ANN) technique has been extensively used to predict and simulate various phenomena of materials as part of the effort to study the relationships between input and output parameters for complex problems [21–29]. The most important feature of an ANN is that this model does not need a specific equation and only requires sufficient datasets of reliable input–output parameters to solve complex

problems. The ANN models have been used to elucidate unclear complex problems and have been successfully applied to some applications in materials science because the ANN can determine the relationship between independent and dependent variables [21–38]. ANN models have been reportedly used to design alloys and to predict microstructures and mechanical properties depending on the chemical compositions, processing conditions, and microstructural factors in steels [29–38]. Jung et al. [38] reported that the tensile properties of high strength steels could be predicted with high accuracy by using an ANN based on the microstructural parameters. However, the relative importance of microstructural factors in determining the tensile properties of high-strength, low-carbon bainitic steels has not yet been studied.

In this study, the primary objectives were: (1) to predict the tensile properties, namely the yield strength, yield-to-tensile ratio, and uniform elongation, for high-strength, low-carbon bainitic steels based on the fraction of constituents; (2) to calculate the influence of constituent fraction on the yield strength, yield-to-tensile ratio, and uniform elongation individually and in combinations of two, and (3) to estimate the influences of the input parameters (the fraction of constituents) on the output parameters (yield strength, yield-to-tensile ratio, and uniform elongation) quantitatively by calculating the index of relative importance. Therefore, an ANN model was applied to design the high-strength, low-carbon bainitic steels with excellent tensile properties, and the results are discussed from a metallurgical perspective.

## 2. Materials and Methods

### 2.1. Alloy and Microstructure

Some high-strength, low-carbon bainitic steels with various chemical compositions and thermo-mechanical controlled process conditions were utilized in this study, and their detailed chemical compositions and manufacturing conditions were explained in our previous studies [8,9]. The fraction of constituents in the steels was quantitatively measured by electron back-scatter diffraction (EBSD, EDAX-TSL, Digiview-IV, EDAX Inc., Mahwah, NJ, USA) analysis. The specimens for the EBSD analysis were mechanically polished and then electro-polished in a mixed solution of 10% perchloric acid ($HClO_4$) and 90% glacial acetic acid ($CH_3COOH$). The working distance, step size, and acceleration voltage for the EBSD operation were 12 mm, 0.18 μm, and 15 kV, respectively. Orientation imaging microscopy (OIM) analysis 7.0 software (TexSEM Laboratories, Inc., Draper, UT, USA) was used to interpret the EBSD results.

According to the previously reported studies on the high-strength, low-carbon steels [5–8], microstructures fabricated by thermo-mechanical controlled process can be classified into four constituents such as PF (polygonal ferrite), AF (acicular ferrite), GB (granular bainite), and BF (bainitic ferrite) based on the crystallographic and morphological characteristics. Figure 1 shows the EBSD grain boundary map with colorized constituent and misorientation along with the geometrically necessary dislocation (GND) density of PF, AF, GB, and BF in training data sample 6. It can be seen that each constituent has different features such as misorientation, grain size, and GND density. Bainitic microstructures (AF, GB, and BF) had a higher GND density than that of PF and exhibited low-angle boundaries in grains. In this study, the fractions of four constituents were used as input parameters of the ANN model to predict the tensile properties of yield strength, yield-to-tensile ratio, and uniform elongation because they act as a key factor for affecting the mechanical properties of low-carbon, high-strength bainitic steels [5–9].

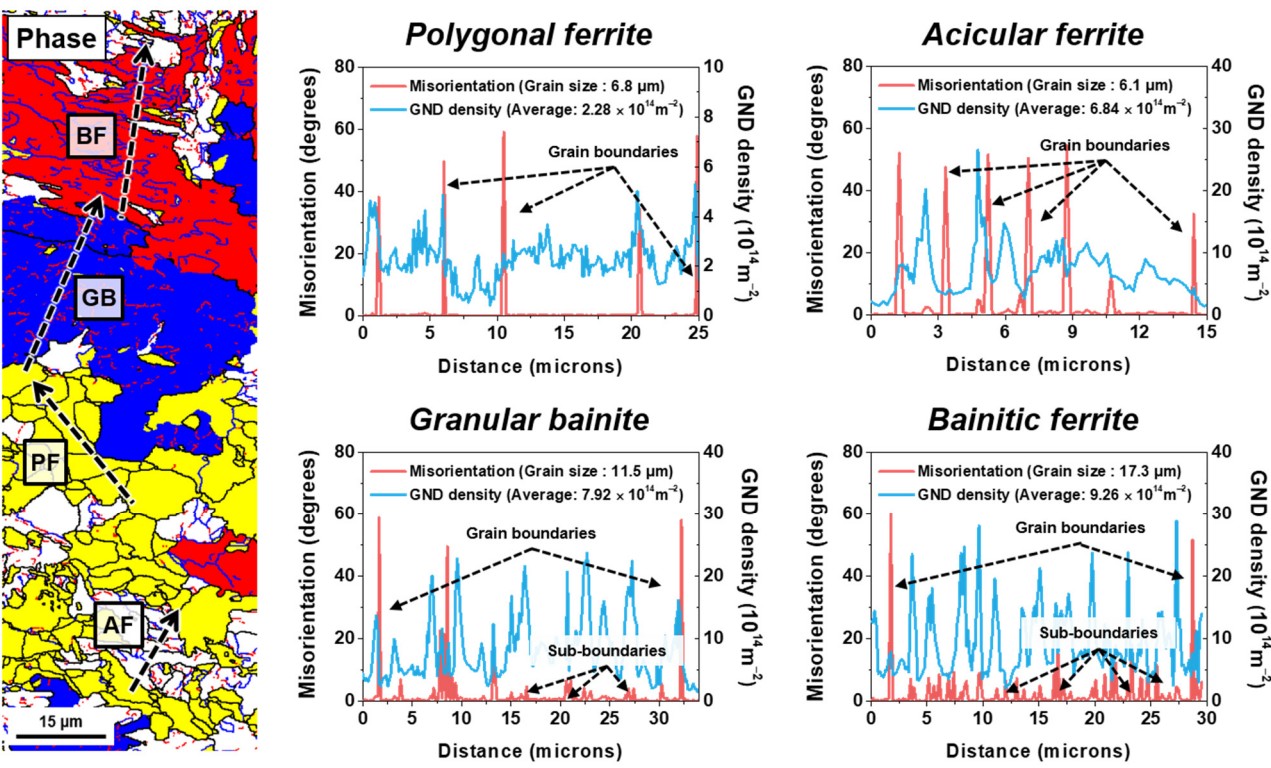

**Figure 1.** EBSD grain boundary map with colorized constituent and misorientation along with the geometrically necessary dislocation (GND) density of the polygonal ferrite, acicular ferrite, granular bainite, and bainitic ferrite in the training data sample 6.

Sub-size round-type tensile specimens with a gage diameter of 6.3 mm and a gage length of 25.4 mm were machined from half-thickness of rolled plates along the longitudinal direction. The room-temperature tensile tests were conducted using a 10-ton universal testing machine (UT-100E, MTDI, Daejeon, Korea) at a constant crosshead speed of 5 mm/min. From the previous stress–strain curves of our tensile tests [9], the tensile properties of the yield strength, yield-to-tensile ratio, and uniform elongation were measured based on the ASTM E8 standard test method [39].

### 2.2. Modeling Establishment

The prediction of tensile properties of low-carbon, high-strength bainitic steels according to the fractions of microstructure constituents was analyzed using the ANN and multi linear regression (MLR) models. The experimental data used in this study consisted of the fraction of PF, AF, GB, and BF constituents with their respective yield strengths, yield-to-tensile ratios, and uniform elongations. From the total of 25 datasets available, 20 datasets were used for model development, and the remaining 5 datasets were kept separate to validate performance of the model. The detailed information of the MLR model was presented in Table S1. For the ANN modeling, all variables were normalized between 0.1 and 0.9. The normalization process is represented quantitatively as follows [26,31]:

$$x_n = \frac{(x - x_{min}) * 0.8}{(x_{max} - x_{min})} + 0.1 \tag{1}$$

where $x_n$ is the normalized value of $x$, and $x_{max}$ and $x_{min}$ are the maximum and minimum values of $x$, respectively, in the entire datasets. Once the best-trained network was found, all of the transformed data were returned to their original values using the following equation [26,31]:

$$x = \frac{(x_n - 0.1)(x_{max} - x_{min})}{0.8} + x_{min} \tag{2}$$

### 2.3. Modeling Procedure

In this study, a back-propagation algorithm and the sigmoid function were used for the development of the ANN. The training program of the present ANN model was written in C [26,31,32]. The ANN model consists of four neurons (the fraction of PF, AF, GB, and BF constituents) in the input layer and three neurons (yield strength, yield-to-tensile ratio, and uniform elongation) in the output layer as shown in Figure 2.

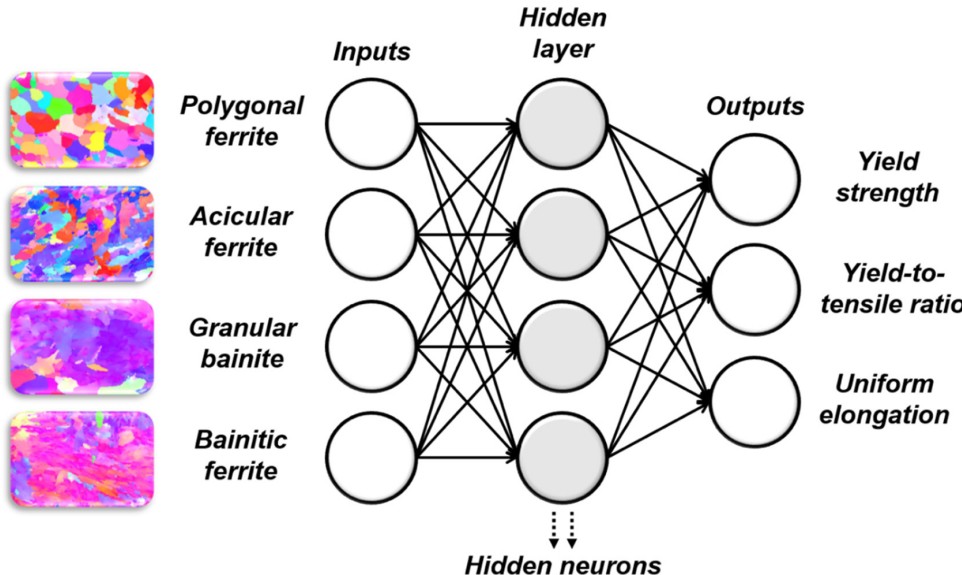

**Figure 2.** Schematic illustration representing the present artificial neural network architecture.

The training of ANN involved the process of adjusting the weights associated with each connection between the neurons until the computed outputs for each set of input data were as close as possible to the experimental output values. The available 25 datasets were divided into 20 training and 5 testing datasets to establish the optimum architecture and to determine the reliability of the developed ANN model (Table 1).

**Table 1.** Training and testing datasets for the artificial neural network modeling of the high-strength, low-carbon bainitic steels.

| Sample | Number | Reference | Constituent Fraction (%) | | | | Tensile Properties | | |
|---|---|---|---|---|---|---|---|---|---|
| | | | Polygonal Ferrite | Acicular Ferrite | Granular Bainite | Bainitic Ferrite | Yield Strength (MPa) | Yield-to-Tensile Ratio | Uniform Elongation (%) |
| Training datasets | 1 | In this study | 8.6 | 57.3 | 19.1 | 15.5 | 448 | 0.80 | 10.8 |
| | 2 | | 4.2 | 77.5 | 9.6 | 8.7 | 499 | 0.82 | 8.9 |
| | 3 | | 9.1 | 58.6 | 20.2 | 12.2 | 578 | 0.84 | 9.7 |
| | 4 | | 5.1 | 55.9 | 11.2 | 27.9 | 500 | 0.85 | 9.3 |
| | 5 | | 12.8 | 35.1 | 28.7 | 23.5 | 484 | 0.84 | 9.8 |
| | 6 | | 26.2 | 49.5 | 12.5 | 11.9 | 480 | 0.86 | 11.2 |
| | 7 | [5] | 54.8 | 35.6 | 0.0 | 8.1 | 500 | 0.86 | 9.9 |
| | 8 | | 61.2 | 23.5 | 0.0 | 10.7 | 460 | 0.88 | 13.1 |
| | 9 | | 38.2 | 40.6 | 0.0 | 17.0 | 510 | 0.89 | 7.6 |
| | 10 | | 80.7 | 12.4 | 0.0 | 4.1 | 437 | 0.89 | 16.6 |
| | 11 | | 70.1 | 20.4 | 0.0 | 6.5 | 430 | 0.87 | 12.4 |
| | 12 | | 74.2 | 18.8 | 0.0 | 1.5 | 429 | 0.86 | 15.1 |
| | 13 | [6] | 0.0 | 8.0 | 87.0 | 5.0 | 612 | 0.72 | 7.6 |
| | 14 | | 0.0 | 18.0 | 9.0 | 73.0 | 728 | 0.74 | 6.5 |
| | 15 | | 0.0 | 76.0 | 18.0 | 6.0 | 626 | 0.78 | 7.1 |
| | 16 | | 0.0 | 11.0 | 78.0 | 11.0 | 608 | 0.71 | 6.8 |
| | 17 | | 0.0 | 87.0 | 12.0 | 2.0 | 641 | 0.81 | 7.8 |
| | 18 | [7] | 70.6 | 3.5 | 0.0 | 23.8 | 459 | 0.88 | 14.0 |
| | 19 | | 19.3 | 57.9 | 8.1 | 11.2 | 607 | 0.85 | 6.9 |
| | 20 | | 24.2 | 52.8 | 4.8 | 14.1 | 597 | 0.88 | 8.0 |

**Table 1.** *Cont.*

| Sample | Number | Reference | Constituent Fraction (%) | | | | Tensile Properties | | |
|---|---|---|---|---|---|---|---|---|---|
| | | | Polygonal Ferrite | Acicular Ferrite | Granular Bainite | Bainitic Ferrite | Yield Strength (MPa) | Yield-to-Tensile Ratio | Uniform Elongation (%) |
| Testing datasets | 1 | In this study | 7.2 | 71.5 | 15.8 | 5.6 | 675 | 0.87 | 8.7 |
| | 2 | [5] | 56.3 | 28.4 | 0.0 | 13.2 | 489 | 0.88 | 12.3 |
| | 3 | | 70.9 | 10.6 | 0.0 | 14.6 | 456 | 0.88 | 12.8 |
| | 4 | [6] | 0.0 | 10.0 | 84.0 | 7.0 | 617 | 0.72 | 7.0 |
| | 5 | [7] | 75.2 | 18.6 | 0.0 | 4.9 | 429 | 0.87 | 14.0 |

The ANN model consists of hidden layers and neurons, a momentum term, a learning rate, and the number of iterations. Throughout the ANN training course, the optimal conditions of the network were determined by the mean error in the predicted output ($E_{rr}$) of the trained data. It can be expressed as follows [27]:

$$E_{rr}(\mathrm{y}) = \frac{1}{N} \sum_{i=1}^{N} |(T_i(y) - O_i(y))| \tag{3}$$

where $N$ is the number of datasets, $T_i$ is the targeted output, and $O_i$ is the calculated output. In this study, the condition with the minimum average error for all output parameters was set as the best condition for training the ANN model. However, if the condition with the minimum average error is different for each output parameter, the condition having the smallest average value of average errors of all output parameters obtained at given condition was set as the best condition. The learning rate, momentum term, and iterations were initially set to 0.6, 0.4, and 5000, respectively. Figure 3 shows the variation of average error as a function of the hidden layers and neurons for the yield strength, yield-to-tensile ratio, and uniform elongation. The minimum average error of the output parameters was achieved with three hidden layers with 100 neurons in each layer. This was selected to find the best condition of the other parameters.

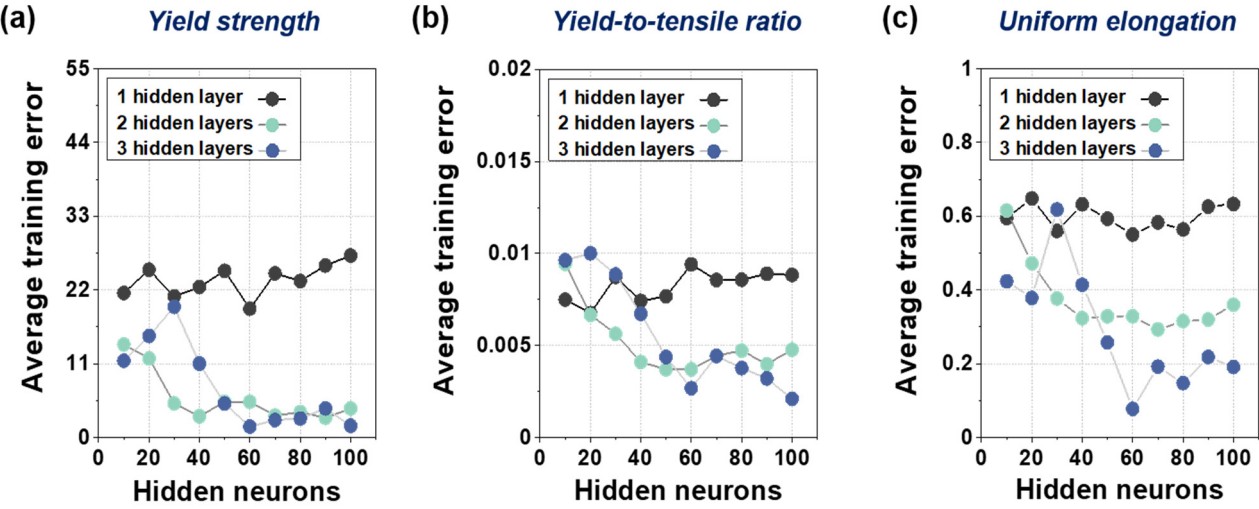

**Figure 3.** Variation of average training errors as a function of hidden layer and neuron for the (**a**) yield strength, (**b**) yield-to-tensile ratio, and (**c**) uniform elongation.

Figure 4 shows the variation of the average training error as a function of the iterations, the learning rate, and the momentum term for the yield strength, yield-to-tensile ratio, and uniform elongation. The number of iterations executed varied from 1000 to 30,000 as shown in Figure 4a to Figure 4c. After 20,000 iterations, the average errors of the yield strength, yield-to-tensile ratio, and uniform elongation were saturated at 0.337807, 0.000141,

and 0.006574, respectively. Hence, the number of iterations was fixed at 20,000. Once the hidden layers, neurons, and iterations were selected, the learning rate and momentum term were varied from 0.1 to 0.9 in steps of 0.1. The minimum average error was obtained at the conditions of a learning rate of 0.5 and a momentum term of 0.4 (Figure 4d–f).

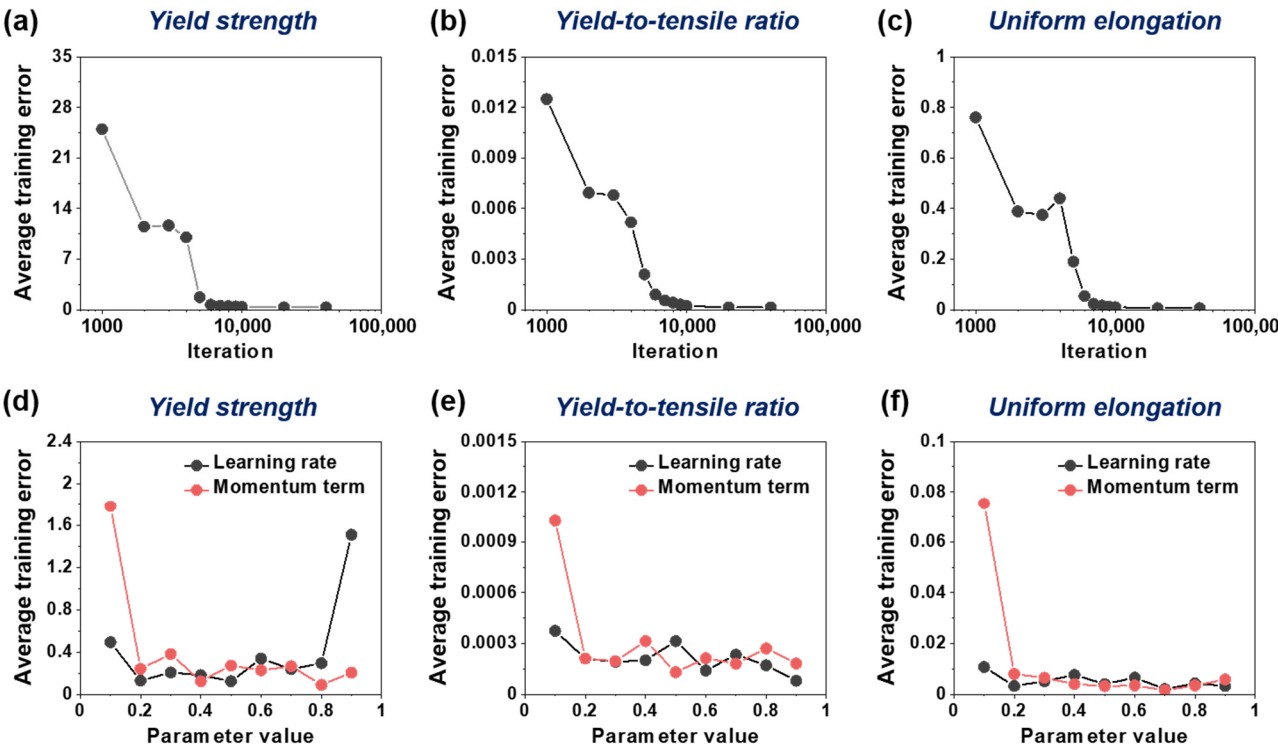

**Figure 4.** Variation of average training errors as a function of the (**a**–**c**) iteration and (**d**–**f**) learning rate and momentum term for the (**a**,**d**) yield strength, (**b**,**e**) yield-to-tensile ratio, and (**c**,**f**) uniform elongation.

## 3. Results and Discussion

### 3.1. Artificial Neural Network Model for Predicting Tensile Properties

In total datasets, as noted above, 20 randomly selected datasets were used for training, and the remaining 5 datasets were used for testing. Figure 5 indicates a comparison of the abilities of the developed ANN and MLR models to predict the yield strength, yield-to-tensile ratio, and uniform elongation of the training datasets in high-strength, low-carbon bainitic steels. The $R^2$ values of the tensile properties for the ANN and MLR models are presented in Figure 5. The tensile properties predicted by ANN model had high $R^2$ values of approximately 1.00, while the results predicted by MLR model indicated relatively lower accuracies despite the training data.

An accurate ANN model is required to generalize the relationships between the input and output parameters for conditions other than those for which the model was trained. It is known that training and testing data affect the construction and performance of model algorithms [28,29]. Therefore, unseen (i.e., previously unemployed datasets during model analysis) testing datasets were used to assess the performance capabilities of the ANN and MLR models. Figure 6 shows the comparison of experimental and predicted tensile properties, and percent error in prediction of the tensile properties of the developed artificial neural network (ANN) and multi linear regression (MLR) models for the testing datasets of high-strength, low-carbon bainitic steels. The respective average percent errors for predicted tensile properties of the testing datasets were indicated in parentheses in Figure 6d–f. It was observed that the ANN model predictions had lower average percent errors for all output parameters compared to those of the MLR model predictions. From these results, it can be seen that the ANN model is in more agreement

with the experimental datasets than the MLR model. Therefore, the present ANN model could predict the tensile properties of the yield strength, yield-to-tensile ratio, and uniform elongation in high-strength, low-carbon bainitic steels as a function of constituent fraction with remarkable accuracy.

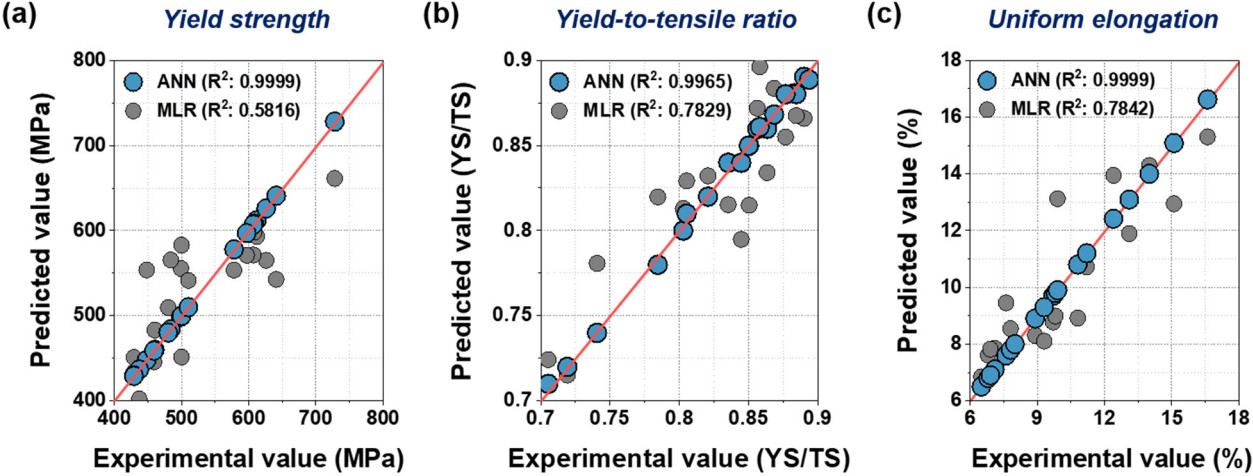

**Figure 5.** Comparison of prediction ability of the developed artificial neural network (ANN) and multi linear regression (MLR) models of high-strength, low-carbon bainitic steels: (**a**) yield strength, (**b**) yield-to-tensile ratio, and (**c**) uniform elongation of the training datasets.

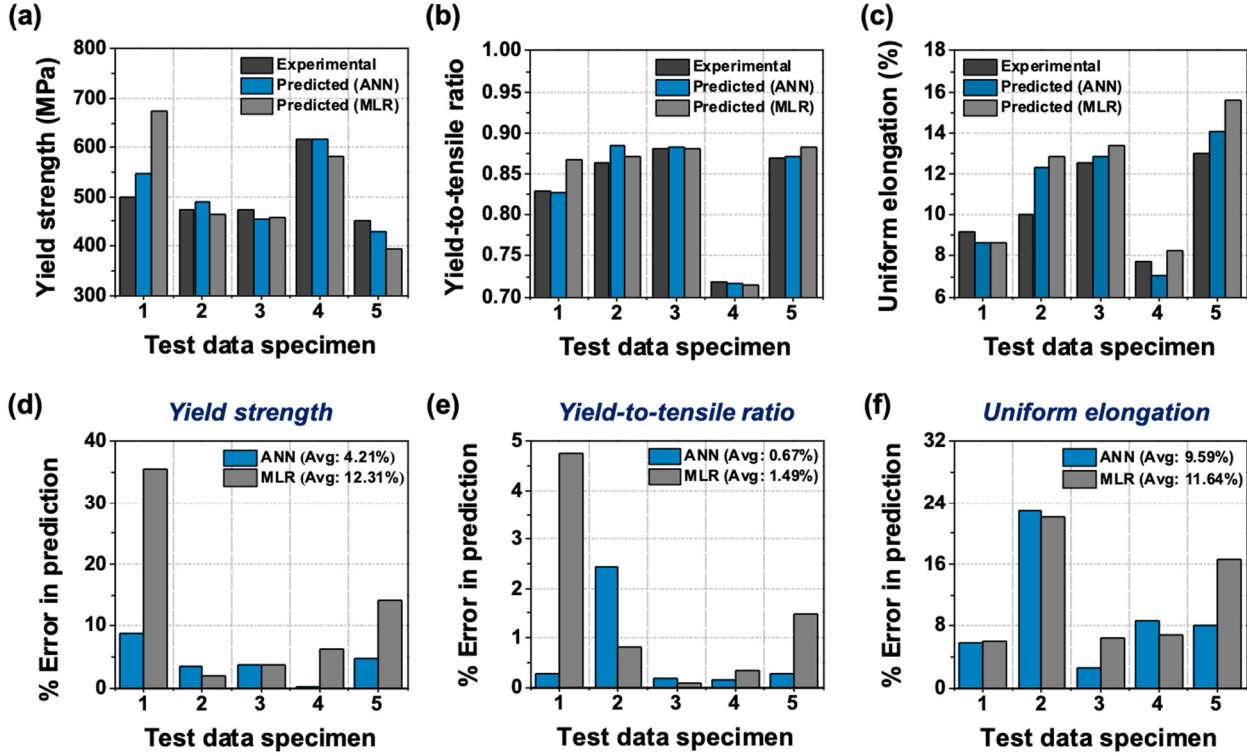

**Figure 6.** Comparison of (**a–c**) experimental and predicted tensile properties, and (**d–f**) percent error in prediction of the tensile properties of the developed artificial neural network (ANN) and multi linear regression (MLR) models for the testing datasets of high-strength, low-carbon bainitic steels. The respective average percent errors for predicted tensile properties of the testing datasets were indicated in parentheses in Figure 6d–f.

### 3.2. Influence of Constituent Fraction on the Tensile Properties Using Sensitivity Analysis

The ANN model with remarkable accuracy for predicting the tensile properties of low-carbon, high-strength bainitic steels based on the fraction of microstructure constituents could be developed in this study. To investigate the sensitivity of input parameters on output parameters in the developed ANN model, the effect of constituent fraction on the yield strength, yield-to-tensile ratio, and uniform elongation individually and in combinations of two was conducted. Figure 7 presents the predicted yield strength, yield-to-tensile ratio, and uniform elongation of the samples with the lowest yield strength, the highest yield-to-tensile ratio, and the lowest uniform elongation, respectively, as a function of the fraction of PF, AF, GB, and BF constituents, keeping a fraction of other constituents. In order to analyze the effect of only one input parameter on output parameters, the values of the other input parameters were fixed. The AF and BF fractions had a greater influence on the variation in the yield strength than the other input parameters. Increasing the AF fraction increased the yield strength, whereas the yield strength decreased again after reaching the maximum value as the BF fraction was increased. Regarding the yield-to-tensile ratio (Figure 7b), it was observed that the yield-to-tensile ratio decreased with an increase in the GB fraction and a decrease in the PF fraction. In contrast, the uniform elongation was mainly affected by the PF and BF fractions and tended to increase with an increase in the PF fraction (Figure 7c).

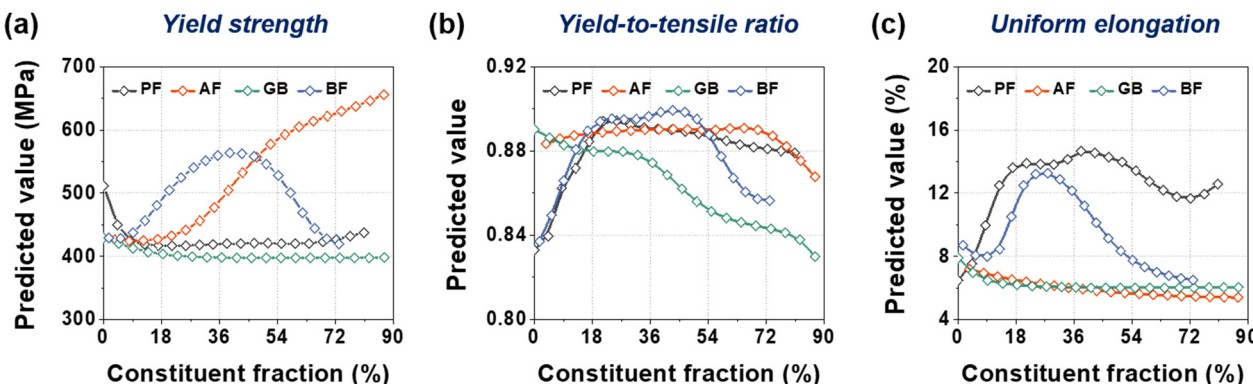

**Figure 7.** Predicted (**a**) yield strength, (**b**) yield-to-tensile ratio, and (**c**) uniform elongation of the samples with the lowest yield strength, the highest yield-to-tensile ratio, and the lowest uniform elongation, respectively, as a function of the fraction of polygonal ferrite, acicular ferrite, granular bainite, and bainitic ferrite constituents, keeping fraction of other constituents.

Based on the results in Figure 7, which show the effects of a single input parameter on the output parameters, the effects of multiple input parameters on the yield strength, yield-to-tensile ratio, and uniform elongation were analyzed and summarized in Figure 8. The contour map allows the visual inspection to select the desired tensile properties. Figure 8a shows the predicted yield strength as a function of the variation in the fractions of the PF and GB at fractions of 43.1% AF and 39.1% BF. The yield strength tended to increase with a decrease in the PF fraction and with an increase in the GB fraction. The predicted yield-to-tensile ratio as a function of the variation in the fractions of PF and AF at fractions of 45.8% GB and 1.5% BF is shown in Figure 8b. It can be seen that the yield-to-tensile ratio decreased with an increase in the AF fraction and a decrease in the PF fraction. Meanwhile, as shown in Figure 8c, which presents the predicted uniform elongation as a function of the variation in the fractions of AF and GB at fractions of 38.2% PF and 27.8% BF, the uniform elongation tended to increase as the AF and GB fractions decreased. As a result, these maps of the predicted tensile properties will provide insight into the constituent fraction needed to make the desired high-strength, low-carbon steel with higher deformability.

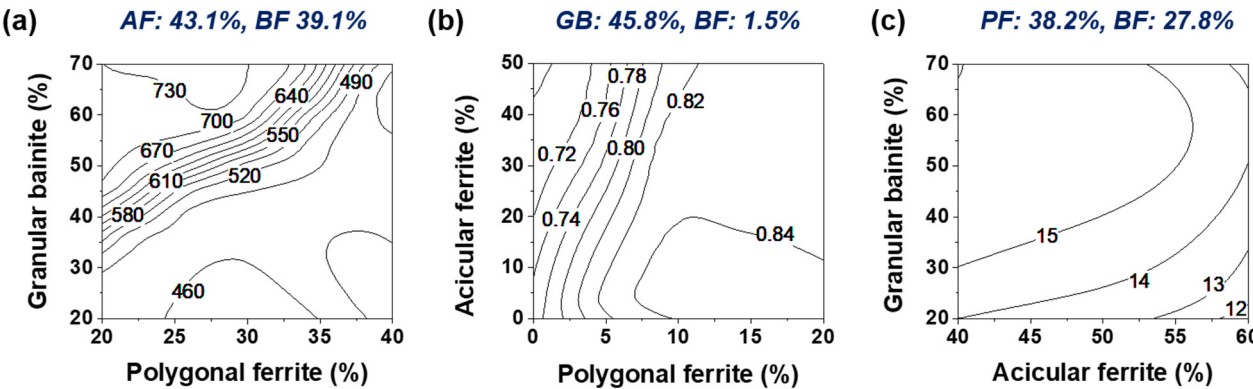

**Figure 8.** Effects of multiple input parameters on the predicted (**a**) yield strength, (**b**) yield-to-tensile ratio, and (**c**) uniform elongation obtained by the developed artificial neural network (ANN) model, keeping fraction of other constituents.

### 3.3. Index of Relative Importance for the Tensile Properties

The index of relative importance ($I_{RI}$) is a vector quantity [31]. The direction and norm of the $I_{RI}$ imply the importance of the input parameters with regard to output parameters. In the present study, the instantaneous importance of the input parameters on the output parameters was evaluated [31]. In the equation $Y = f(X_1 + X_2 + X_3 + X_4)$, Y is the yield strength, yield-to-tensile ratio, or uniform elongation; and $X_1$ to $X_4$ represent the fraction of the PF, AF, GB, and BF constituents, respectively. The process of calculating the $I_{RI}$ is shown in Table S2.

Figure 9 shows the average $I_{RI}$ of the input parameters of the PF, AF, GB, and BF fractions for the yield strength, yield-to-tensile ratio, and uniform elongation for all datasets. For the average $I_{RI}$ for the yield strength (Figure 9a), PF, AF, and GB had positive effects, while BF indicated a negative effect. In particular, it was found that increasing the AF fraction was effective in improving the yield strength. With regard to the yield-to-tensile ratio (Figure 9b), the PF and bainitic microstructures (AF, GB, and BF) had the opposite effect, and the yield-to-tensile ratio decreased with an increase in the fraction of the bainitic microstructures (AF, GB, and BF). Meanwhile, as shown in Figure 9c, the PF and GB had a positive effect, whereas the AF and BF exhibited negative effects on the uniform elongation. In particular, the PF had a strong effect on the uniform elongation compared to the other input parameters.

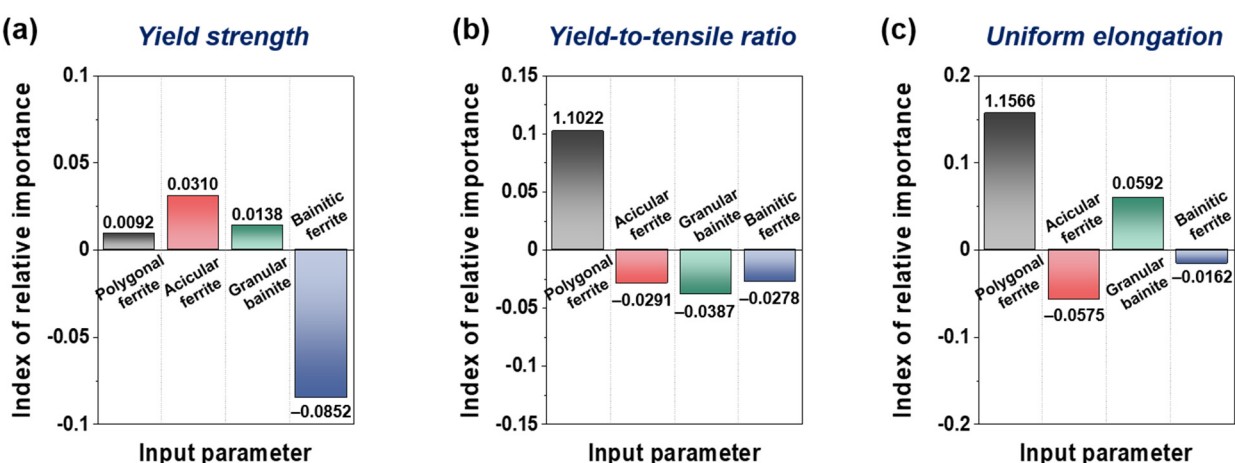

**Figure 9.** Average index of relative importance of input parameters of fraction of polygonal ferrite, acicular ferrite, granular bainite, and bainitic ferrite constituents on the (**a**) yield strength, (**b**) yield-to-tensile ratio, and (**c**) uniform elongation for all datasets.

The variation in the tensile properties according to the input parameters can be explained by the microstructural characteristics of each constituent. As shown in Figure 1, the AF had irregularly shaped grains with high-angle boundaries and exhibited the finest grain size, while the BF had the largest grain size with low-angle boundaries. It is well known that the yield strength increases when the dislocation slips are inhibited by many microstructural factors, such as the grain boundary, solute atoms, precipitates, stress field, and dislocations [40–42]. Accordingly, it is reasonable to conclude that the yield strength can be improved by increasing the fraction of AF with the finest grain size due to the grain boundary strengthening effect.

Meanwhile, the yield-to-tensile ratio is related to the work hardening after yielding [40]. Lee et al. [43] reported that bainitic microstructures (AF, GB, and BF) show higher work hardening ability due to their high dislocation densities. As mentioned previously, the bainitic microstructures (AF, GB, and BF) have a negative effect on the yield-to-tensile ratio (Figure 9b), and it can increase the work hardening by inhibiting dislocation slips. Therefore, it is understood that the bainitic microstructures (AF, GB, and BF) exhibit the greatest effect on decreasing the yield-to-tensile ratio owing to their higher work hardening ability. With it respect to uniform elongation (Figure 9c), it is known that the uniform elongation can be improved when dislocation slips readily occur inside the grains [43]. The PF causes dislocations to move easily given that it is fully recrystallized and has the lowest dislocation density (Figure 1). Accordingly, it is reasonable to consider that high-strength, low-carbon steel with a higher fraction of PF has a relatively higher uniform elongation due to the easily occurring dislocation slips in the PF.

Consequently, the tensile properties such as the yield strength, yield-to-tensile ratio, and uniform elongation of low-carbon, high-strength bainitic steels could be predicted by adopting the ANN model with high accuracy based on the fraction of microstructure constituents. These findings represent that the fraction of microstructure constituents is a key factor for determining the tensile properties in low-carbon, high-strength bainitic steels. As a result, the application of the ANN model can be highly utilized in designing an optimal microstructure to obtain desired tensile properties in given various environments.

## 4. Conclusions

Based on the present investigation of the application of ANN to the prediction of the tensile properties in high-strength, low-carbon bainitic steels based on the fraction of constituents, the following conclusions can be drawn.

1. An ANN model that enables predictions of the yield strength, yield-to-tensile ratio, and uniform elongation as a function of the fraction of constituents such as PF, AF, GB, and BF was developed. The prediction of the tensile properties made using the ANN model was more accurate than that using the MLR model.

2. The results of the variation in output parameters according to the one input parameter indicated that the yield strength, yield-to-tensile ratio, and uniform elongation were changed mainly by the fractions of AF, GB, and PF. From the effect of multiple input parameters, a microstructure concept would be suggested to make the desired high-strength, low-carbon steel with higher deformability.

3. Based on the average index of the relative importance ($I_{RI}$) for the input parameters, the yield strength, yield-to-tensile ratio, and uniform elongation were effectively improved by increasing the fraction of AF, bainitic microstructures (AF, GB, and BF), and PF, respectively, in terms of the work hardening and dislocation slip behavior according to the microstructural features, such as the grain size and dislocation density.

**Supplementary Materials:** The following are available online at https://www.mdpi.com/article/10.3390/met11081314/s1, Table S1: Multi linear regression (MLR) model for predicting the yield strength, yield-to-tensile ratio, and uniform elongation of low-carbon high-strength bainitic steels based on the fractions of microstructure constituents. Table S2: Process to calculate the index of relative importance ($I_{RI}$).

**Author Contributions:** Conceptualization, B.H.; formal analysis, S.-H.S.; investigation, S.-I.L.; data curation, S.-I.L.; writing—original draft preparation, S.-I.L.; writing—review and editing, B.H., S.-H.S. All authors have read and agreed to the published version of the manuscript.

**Funding:** This research was supported by the Technology Innovation Program (Grant No. 20015945) funded by the Ministry of Trade, Industry and Energy (MOTIE) and the Basic Science Research Program through the National Research Foundation of Korea (NRF-2017R1A2B2009336).

**Institutional Review Board Statement:** Not applicable.

**Informed Consent Statement:** Not applicable.

**Data Availability Statement:** Not applicable.

**Acknowledgments:** The authors would like to thank N.S. Reddy of Gyeongsang National University, Chan Hee Park and P.L. Narayana of Korea Institute of Materials Science for the instructions of artificial neural network program.

**Conflicts of Interest:** The authors declare no conflict of interest.

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
