# Peer review of "Application of Artificial Neural Network to the Prediction of Tensile Properties in High-Strength Low-Carbon Bainitic Steels"

_metals, doi:10.3390/met11081314_

Round 1
Reviewer 1 Report
- Without feature preprocessing and feature selection, PF, AF, GB and BF are directly selected as feature input, is it not rigorous enough?
- There are only 25 article data sets. Do the 25 data sets in the article mean 25 data sets? Does such a small data set lead to overfitting problems? Will it affect the generalization ability of the model?
- After obtaining the data, should the data be processed and checked to see how the data is distributed and whether the data is representative?
- What is the basis for data standardization between 0.1 and 0.9, and why is it standardized?
- Does the use of a backpropagation algorithm refer to the BP neural network?
Should we explain the reason for using Sigmoid function as activation function directly?
- In this study, the initial values of learning rate, momentum term and iteration number are 0.6, 0.4 and 5000, respectively. What is the basis for setting this initial value and what is its special significance are not explained.
- In this paper, it is pointed out that as close as possible to the experimental data, should the problem of model overfitting be considered?
- There are many indicators of machine learning evaluation and prediction model. Why choose average error and compare it with other indicators?
- It is said that the R2 value of tensile properties predicted by neural network model is relatively high, which is about 0.99, and corresponding to Figure 5, will there be over-fitting phenomenon, and the practicability of the model needs to be verified?
- The title of this paper mentions sensitivity analysis method to study the influence of components on tensile properties, which should be followed by an explanation of what sensitivity analysis method is.
- When discussing the single input parameter, the article said:Increasing the AF fraction increased the yield strength, whereas the yield strength decreased again after reaching the maximum value as the BF fraction was increased. Regarding the yield-to-tensile ratio , it was observed that the yield-to-tensile ratio decreased with an increase in the GB fraction and a decrease in the PF fraction. However, when discussing several input parameters later, it is concluded thatit was found that the PF and bainitic microstructures (AF, GB, and BF) had opposite effects on the yield strength, yield-to-tensile ratio, and uniform elongation.
There are contradictions between the two conclusions here, and there is no explanation here why this happens.
- In Figure 9a, BF has obvious negative effect on yield strength, and why this result appears is not discussed in the article. Similarly, in Figure 9b, PF has a strong positive effect on yield-tensile ratio, and no corresponding explanation is given.
Reviewer 2 Report
General Comment
The manuscript presents an artificial neural network (ANN) model to predict the tensile properties (yield strength, yield-to-tensile ratio and uniform elongation), as output parameters, of high-strength low-carbon bainitic steels, as function of the fraction of constituents, as input parameters, namely: PF (polygonal ferrite), acicular ferrite (AF), granular bainite (GB) and bainitic ferrite (BF). For this, in addition to data from the literature, additional low-carbon bainitic steels were fabricated and tested. The collected results were used as training datasets and testing datasets for the proposed ANN model. The modelling procedure and validation of the ANN model is described and discussed. Then, using a sensitivity analysis, the ANN model is used to study the influence of constituent fractions on the tensile properties of high-strength low-carbon bainitic steels. The obtained results are presented and discussed.
The topic of the manuscript is very interesting and actual since it shows how ANN can be used as a reliable tool to predict the mechanical properties of materials. The proposed ANN can provide a deeper understanding of the relationships between constituent fraction and tensile properties of low-carbon bainitic steels, and help to fabric steels with higher mechanical performance. The results of this study can be useful for the metallurgic industry.
I made some comments to improve the manuscript and also asked for some clarifications. The authors should take the comments into account and revise their manuscript.
Specific Comment 1
Introduction
Being a research article, the introduction section is too short and very poor. It is not correct to write sentences ended by several references between brackets. Please avoid this! The discussion of the literature review and main obtained results must be deeply improved in relation with the topic of the manuscript. Namely, the correlation between the microstructure and tensile properties of steel, as well as the use of ANN models to design steel alloys and predict microstructures and mechanical properties, must be deeply discussed, in order to justify the need and novelty of your study.
Specific Comment 2
Introduction, line 55
The sentence “…(1) to predict the tensile properties of the yield strength, yield-to-tensile ratio, and uniform elongation in high-strength …” must be reviewed. For instance, it would be better to write “…(1) to predict the tensile properties, namely the yield strength, yield-to-tensile ratio and uniform elongation, for high-strength …”.
Specific Comment 3
Section 2.1
It is stated that some high-strength low-carbon bainitic steels were fabricated for this study. However, the given information is very scarce. Please provide much more information and details on the used controlled process conditions and obtained alloys.
Specific Comment 4
Section 2.1
Please give examples of the stress-strain curves obtained through the performed tensile tests on the fabricated steel samples. Also, define, explain and illustrate how the relevant mechanical properties (yield strength, yield-to-tensile ratio and uniform elongation) where obtained from such curves.
Specific Comment 5
Section 2.3
Please provide more details about the computational implementation of the ANN model. For instance, present and explain the algorithm for the calculation procedure.
Specific Comment 6
Section 2.3
Please provide details about the used multi linear regression (MLR) model, as for the ANN model.
Specific Comment 7
Table 1
Please clarify the unit for the uniform elongation. It should be % or ‰.
Specific Comment 8
Page 5 and Fig. 3
It is stated that “The minimum average error of the output parameters was achieved with three hidden layers with 100 neurons in each layer”. From Fig. 3 this is not clear! For 60 neurons, the average error is also very low (yield strength and yield-to-tensile ratio) or even smaller (uniform elongation). Please clarify this better in the manuscript.
Specific Comment 9
Page 6, line 147
Avoid to write sentences in the first-person, such as at line 147 (“we determined…”).
Specific Comment 10
Page 6 and Fig. 4
It is stated that “We determined that a learning rate of 0.5 and a momentum term of 0.4 generated minimum errors”. This seems not to be entirely true. For instance, in Fig. 4(e), the average training error for the learning rate is lower for 0.9 and for the momentum term is lower for 0.5. Please clarify this better in the manuscript.
Specific Comment 11
Section 3.1, line 160
Based on the values presented for R^2 in Fig 5, by rounding it should be “1.00” and not “0.99”
Specific Comment 12
Fig. 6
Please present also the results in term of the experimental and predicted values of the tensile properties (yield strength, yield-to-tensile ratio and uniform elongation) for visual comparison.
Specific Comment 13
Section 3.2
The first sentence should start clarifying that the performed study in this section was made by using the ANN model.
Also, explain what do you mean by “”with the worst value of each property” and also “keeping fraction of other constituents”. Please clarify this better in the manuscript.
Specific Comment 14
Section 3.2 + Fig. 7
In some of the presented curves and for some of the tensile properties, local maxima can be visualized (the predicted value increases and then it decreases). Please give possible physical explanations for this observed behavior.
Specific Comment 15
Section 3.3 + Line 224/225
Since I_RI is a vector quantity, it would be better to write “norm” instead of “amount”
Specific Comment 16
Conclusions + Figure 9
The conclusion stated at point 2 is nor entirely true for the uniform elongation (Fig. 9(c)). Please clarify better in the manuscript.
Author Response
Please see the attached response file.

Round 2
Reviewer 1 Report
- The process of calculating the IRI is shown in Table S2.The table S2 mentioned here does not appear.
- The dislocation density and color diagram in Figure 1 are not explained in detail in this paper.
- Can the article explain the social practicability of the model at the end?
- BF has obvious negative effect on yield strength in fig. 9a, and why this result appears is not discussed in the article. Similarly, in Figure 9b, PF has a strong positive effect on yield-tensile ratio, and no corresponding explanation is given.
- The serial numbers of the cited documents should be marked from small to large.
5.The activation function mentioned in the article is sigmoid, does it mean that the activation function of both the hidden layer and the output layer is sigmoid function?
6.When adjusting the number of hidden layers and the number of neurons in each layer, why just control to 3 and 100?
7.The detailed ratio between the predicted value and the test value is not written in the paper.
Author Response
Please find the file attached.

Reviewer 2 Report
Response 1:
The introduction section was not improved as previously requested by the reviewer (previous Specific Comment 1). In their reply, the authors state that:
“There are not many studies reported because these steels had very complex bainitic microstructures which have different features from conventional bainite microstructures. Therefore, we believe that the purpose of the study was enough explained because few studies had been conducted to predict tensile properties of low-carbon high-strength bainitic steels based on the microstructure fraction by applying ANN models within limited data sets:
Hence, a discussion of the main findings of such “few studies” is required to improve the introduction section, in relation with the topic of the submitted article.
Response 3:
Since information of the fabricated high-strength low-carbon bainitic steels was given in previous studies, and references were added, the first sentence of Section 2.1 should not include “… were fabricated in this study.”
Author Response
Please find the file attached.

Round 3
Reviewer 1 Report
- when analyzing the influence of multiple input parameters on yield strength, yield-tensile ratio and uniform elongation, Figure 8a shows the predicted yield strength as a function of the variation in the fractions of the PF and GB at fractions of 43.1 % AF and 39.1 % BF. How are the values of 43.1% and 39.1% determined here, and what is the significance? And the values involved in fig. 8b and fig. 8c.
Author Response
Response to Reviewer 1 Comments
Point 1: When analyzing the influence of multiple input parameters on yield strength, yield-tensile ratio and uniform elongation, Figure 8a shows the predicted yield strength as a function of the variation in the fractions of the PF and GB at fractions of 43.1 % AF and 39.1 % BF. How are the values of 43.1% and 39.1% determined here, and what is the significance? And the values involved in fig. 8b and fig. 8c.
Response 1: The constant values of input parameters, such as 43.1% and 39.1% in Figure 8(a), have no significant meaning. In the present study, the constant values of input parameters were just set as maximum, minimum, and intermediate values in order to simultaneously confirm the effect of the two input parameters on the output parameters by fixing the values of the other two input parameters to specific values. Among total results, the graphs with the representative results of each output parameter were indicated in Figure 8.
Reviewer 2 Report
I´m more satisfied with the authors’ replies to my previous comments and with the change in the introduction section to improve the article. I consider that the article can be accepted in the present form.
Author Response
We are pleased to know that you considered the paper can be accepted in the present form for publication in Metals.